# Quadrupolar excitons and hybridized interlayer Mott insulator in a trilayer moiré superlattice

Zhen Lian[1,9], Dongxue Chen [1,9], Lei Ma [1,9], Yuze Meng[1,9], Ying Su[2], Li Yan[1], Xiong Huang [3,4], Qiran Wu[3], Xinyue Chen[1], Mark Blei[5], Takashi Taniguchi [6], Kenji Watanabe [7], Sefaattin Tongay [5], Chuanwei Zhang [2], Yong-Tao Cui [3] ✉ & Su-Fei Shi [1,8] ✉

Transition metal dichalcogenide (TMDC) moiré superlattices, owing to the moiré flatbands and strong correlation, can host periodic electron crystals and fascinating correlated physics. The TMDC heterojunctions in the type-II alignment also enable long-lived interlayer excitons that are promising for correlated bosonic states, while the interaction is dictated by the asymmetry of the heterojunction. Here we demonstrate a new excitonic state, quadrupolar exciton, in a symmetric $WSe_2$-$WS_2$-$WSe_2$ trilayer moiré superlattice. The quadrupolar excitons exhibit a quadratic dependence on the electric field, distinctively different from the linear Stark shift of the dipolar excitons in heterobilayers. This quadrupolar exciton stems from the hybridization of $WSe_2$ valence moiré flatbands. The same mechanism also gives rise to an interlayer Mott insulator state, in which the two $WSe_2$ layers share one hole laterally confined in one moiré unit cell. In contrast, the hole occupation probability in each layer can be continuously tuned via an out-of-plane electric field, reaching 100% in the top or bottom $WSe_2$ under a large electric field, accompanying the transition from quadrupolar excitons to dipolar excitons. Our work demonstrates a trilayer moiré system as a new exciting playground for realizing novel correlated states and engineering quantum phase transitions.

Monolayer TMDCs, as atomically thin direct bandgap semiconductors, offer a unique playground to explore novel optoelectronic phenomena[1,2], especially with the ability to form heterostructures that enable a new range of control knobs. For example, TMDC heterojunctions in a type-II alignment host long-lived interlayer excitons[3–6],

with electrons and holes residing in different layers[3,4]. These interlayer excitons possess the valley degree of freedom, as well as a large Stark shift due to the permanent dipole moment, rendering them promising candidates as tunable quantum emitters[6]. Recently, angle-aligned TMDC moiré superlattices exhibit strong Coulomb interactions in the

[1]Department of Chemical and Biological Engineering, Rensselaer Polytechnic Institute, Troy, NY 12180, USA. [2]Department of Physics, University of Texas at Dallas, Dallas, TX 75083, USA. [3]Department of Physics and Astronomy, University of California, Riverside, CA 92521, USA. [4]Department of Materials Science and Engineering, University of California, Riverside, CA 92521, USA. [5]School for Engineering of Matter, Transport and Energy, Arizona State University, Tempe, AZ 85287, USA. [6]International Center for Materials Nanoarchitectonics, National Institute for Materials Science, 1-1 Namiki, Tsukuba 305-0044, Japan. [7]Research Center for Functional Materials, National Institute for Materials Science, 1-1 Namiki, Tsukuba 305-0044, Japan. [8]Department of Electrical, Computer & Systems Engineering, Rensselaer Polytechnic Institute, Troy, NY 12180, USA. [9]These authors contributed equally: Zhen Lian, Dongxue Chen, Lei Ma, Yuze Meng. ✉e-mail: yongtao.cui@ucr.edu; shis2@rpi.edu

electronic flatbands, leading to correlated states[7–19] such as Mott insulator and generalized Wigner crystal[7,8,12,17]. The moiré coupling also gives rise to flat excitonic bands[20–23] that could potentially be utilized to realize correlated bosonic states[24], such as Bose-Einstein condensation (BEC) and superfluidity[25–27]. The interaction between interlayer excitons is dominated by the repulsive force between their permanent dipoles, whose alignment is dictated by the asymmetry of the heterostructure, with electrons and holes separated in two different layers.

In this work, we report a new interlayer quadrupolar exciton in a symmetric TMDC heterostructure: angle-aligned WSe₂/WS₂/WSe₂ trilayer. The interlayer excitons in the top and bottom bilayers have opposite polarities, which restores the symmetry. Their hybridization then forms a superposition state of interlayer excitons, canceling the dipolar moments and giving rise to a quadrupolar exciton, which has been predicted to enable intriguing quantum phase transition[26,28–30]. In the presence of moiré coupling, this hybridization further gives rise to a new type of correlated electronic state, hybridized interlayer Mott insulator, in which the correlated holes are shared between the two WSe₂ layers, and the layer population can be continuously tuned by an electric field.

## Results and discussion
### Quadrupolar exciton in angle-aligned t-WSe₂/WS₂/b-WSe₂ trilayer

The typical device structure is schematically shown in Fig. 1a, which contains three regions of different stackings among the three monolayers: (I) WS₂ over the bottom WSe₂, which we denote as WS₂/b-WSe₂; (II) top WSe₂ over WS₂ (t-WSe₂/WS₂); and (III) the t-WSe₂/WS₂/b-WSe₂ trilayer. The whole device is gated by the top and bottom gate

electrodes made of few-layer graphene (FLG), which provides independent control of the electric field and doping.

In the bilayer regions I and II, the WSe₂/WS₂ moiré superlattices host both correlated electrons and interlayer excitons due to the type-II band alignment (Fig. 1c). The interlayer excitons, with holes residing in the WSe₂ layer and electrons in the WS₂ layer (Fig. 1c), interact with the correlated electrons and can be used to read out the transitions at the correlated insulating states[17,31–34]. The doping-dependent photoluminescence (PL) spectra in these regions (Fig. 1e, f) clearly reveal these features: the interlayer exciton PL peak has a strong intensity at the charge neutrality point (CNP), which decreases quickly upon doping; the PL energy and intensity are also modulated by correlated insulator states such as the Mott insulator states at both $n = 1$ and $-1$, consistent with the previous studies[12,31].

In the trilayer region III, we expect quadrupolar excitons as schematically plotted in Fig. 1b. The quadrupolar exciton is the superposition of the two dipolar excitons of opposite polarities through the hybridization of the valence bands in the top and bottom WSe₂ layers, which leads to the splitting of valence bands, $\Delta^{\pm}$, as shown in Fig. 1d, similar to the formation of bonding and antibonding states in a double-well system[29]. As a result, the quadrupolar excitons will have two branches: one at lower energy than the dipolar exciton and the other at higher energy, assuming that all have similar binding energies[29]. Figure 1g plots the PL in this region, which indeed exhibits a major PL resonance at energies below the dipolar excitons in Fig. 1e and f. We have not observed any PL resonance corresponding to the higher energy quadrupolar exciton yet, while some devices show high energy exciton PL with different nature that we are going to explore in the future (details in Supplementary Information Section 18). The doping dependence is also drastically different: the intensity of the

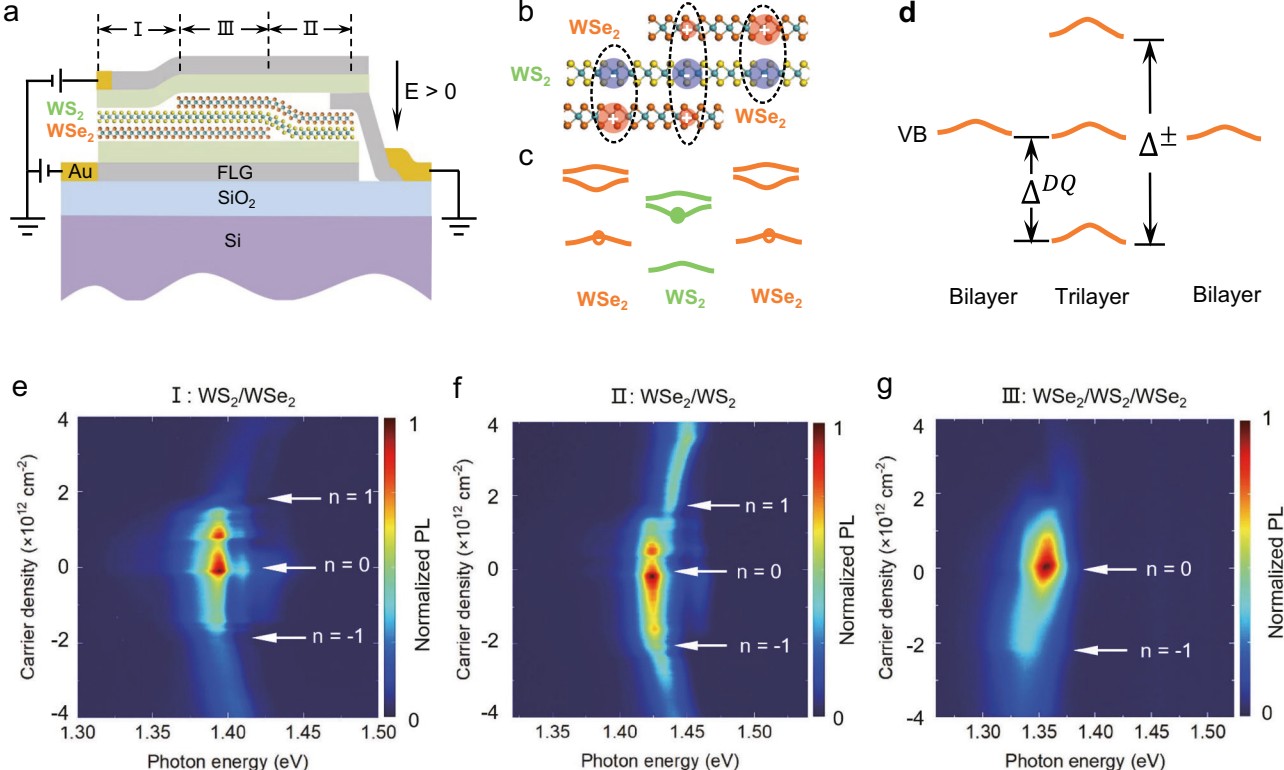

**Fig. 1 | Excitons in different stacking structures of the trilayer device.**
**a** Schematics of the device structure with three different regions: (I) WS₂/b-WSe₂ (II) t-WSe₂/WS₂/b-WSe₂ (III) top bilayer, t-WSe₂/WS₂. **b** Schematics of the dipolar and quadrupolar excitons configuration. **c** Type-II alignment of the angle-aligned WSe₂/ WS₂ heterobilayer. **d** Valence band hybridization in the trilayer region, compared with the flat valence band of WSe₂ in the WSe₂/WS₂ moiré bilayer regions. **e–g** are doping-dependent PL spectra for regions I, II, and III. The PL data were taken from device D5.

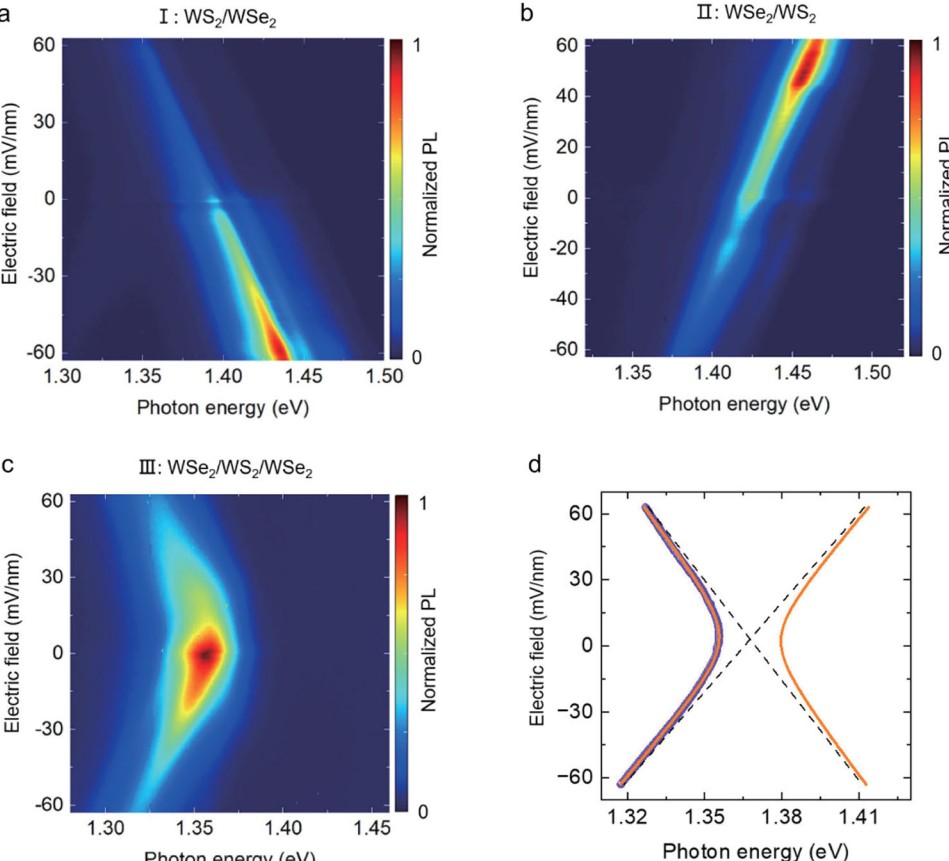

**Fig. 2 | Electric field-dependent PL spectra of dipolar and quadrupolar excitons.** **a**–**c** are electric field-dependent PL spectra of the region I, II, and III, respectively. **d** Fitting of the quadrupolar excitons PL resonances (orange curve) on extracted PL peak energy (purple spheres) as a function of the electric field from (**c**). The PL peak positions are extracted by fitting each PL spectrum with a single Lorentzian peak. The PL data were taken from device D5.

lower energy PL peak retains upon hole doping and only starts to decay at $n = -1$ (we will discuss this in more detail later).

The quadrupole nature of the excitons in the trilayer region is confirmed by the electric field-dependent PL spectra. In regions I (Fig. 2a) and II (Fig. 2b), the interlayer exciton PL peaks both shift linearly as a function of the out-of-plane electric field but with opposite signs of the slope. The slope is $-0.72\ e \cdot nm$ for WS$_2$/b-WSe$_2$ (region I) and $0.66\ e \cdot nm$ for t-WSe$_2$/WS$_2$ (region II), consistent with the previous results[21,35–39]. In contrast, the PL from the trilayer region III is symmetric about the electric field, and the resonance energy exhibits a quadratic dependence on the electric field, as shown in Fig. 2c, d, clearly demonstrating that the trilayer PL is from quadrupolar excitons. The PL resonance energy can be well fitted by a quadrupolar exciton model (orange curves in Fig. 2d, details in Supplementary Information Section 9). It is worth noting that at large electric fields, the quadrupolar exciton approaches the linear Stark shift of dipolar excitons with a slope around $0.7\ e \cdot nm$ (dashed lines), matching what we extracted from the data in the bilayer regions I and II. We further extract the $\Delta^{DQ}$, the energy difference between dipolar excitons and quadrupolar excitons under net zero electric field, to be about 12 meV from the fitting in Fig. 2d (Supplementary Information Section 9), consistent with the theoretical calculation for a similar trilayer structure (10–30 meV in WSe$_2$/MoSe$_2$/WSe$_2$)[29]. We have also reproduced similar quadrupolar exciton behaviors in other angle-aligned WSe$_2$/WS$_2$/WSe$_2$ devices, which show a $\Delta^{DQ}$ about 30 meV (device D2, Supplementary Information Section 10) and 9 meV (device D1 and D3, Supplementary Information Section 14). We note that the dipolar exciton resonance energies in regions I and II only serve as a guide for the two dipolar excitons involved in forming the quadrupolar excitons due to

dielectric environment difference and possible spatial inhomogeneity. The energies of the two dipolar excitons that form quadrupolar excitons in region III can be extracted from the fitting and are similar in values, typically less than 7 meV (detailed discussion in Supplementary Information Section 14). In fact, the electric field dependence of the quadrupolar exciton can be used to extract the energy difference between the two dipolar excitons involved in the hybridization, which dictates the hybridization to occur at a finite electric field that tunes the two dipolar exciton energies into resonance (details in Supplementary Information Section 14). We also want to mention that the higher energy mode of the predicted quadrupolar excitons (asymmetric quadrupolar exciton mode[29]) is missing in Fig. 2, likely due to the excited state or even dark state nature[40] of the quadrupolar exciton, which leads to the absence of PL.

The quadrupolar excitons show distinctively different power dependence compared with that from dipolar excitons, as shown in Fig. S5. The integrated PL intensity of quadrupolar excitons exhibits more nonlinear dependence than dipolar excitons, likely due to their larger size. In addition, the PL peak blueshifts as a function of the excitation power (Fig. S5b, e) or exciton density (Fig. S6) is smaller for quadrupolar exciton compared with that of dipolar excitons, consistent with our expectation of reduced exciton-exciton repulsion for quadrupolar excitons. The estimation of the exciton density can be found in Supplementary Information Section 16.

## Evidence of an interlayer Mott insulator
Next, we study the interaction between the quadrupolar exciton and the correlated electrons in the moiré flatlands. We first revisit the doping dependence of the quadrupolar exciton at zero electric field. Here, the

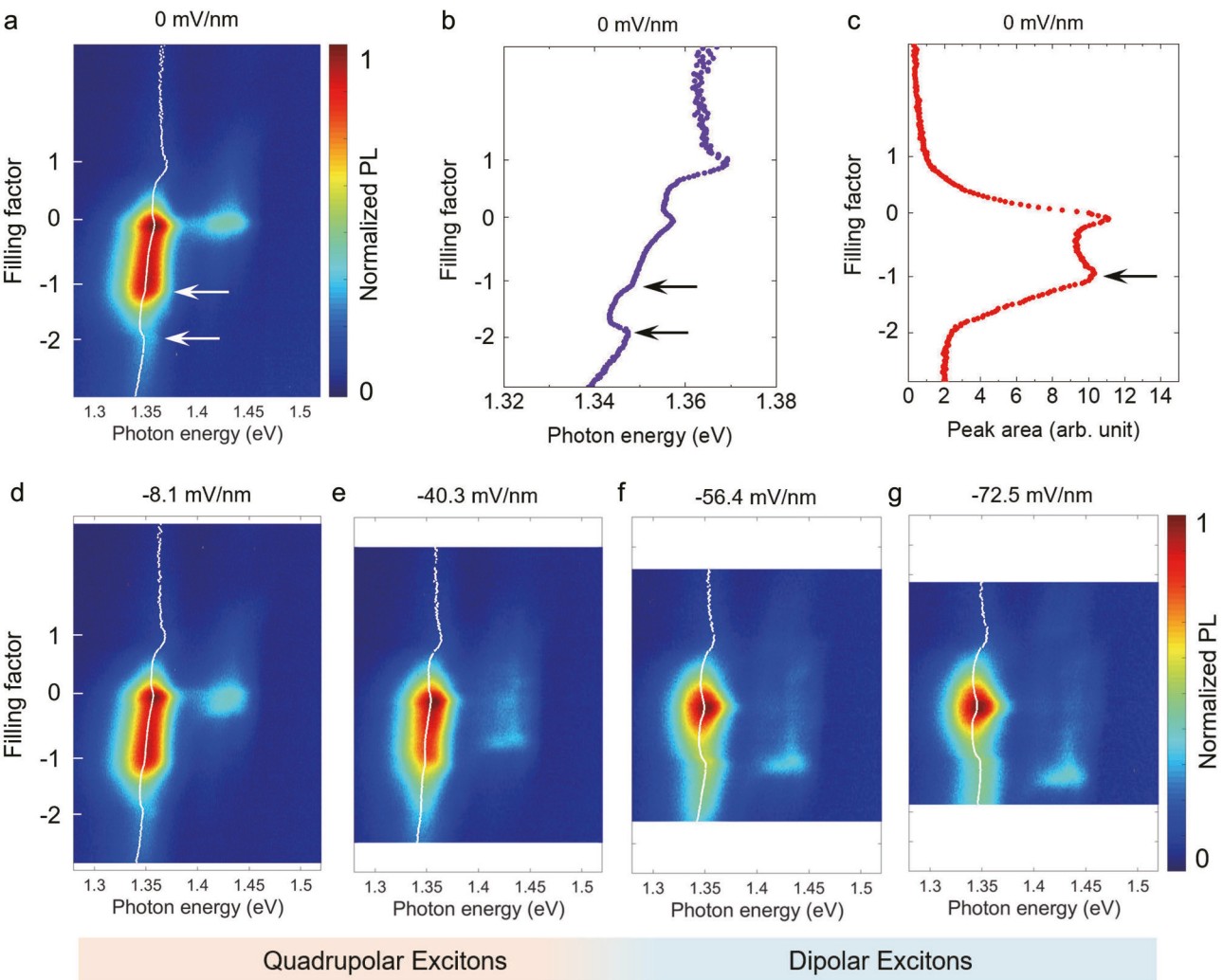

**Fig. 3 | Evolution of doping-dependent PL spectra of the trilayer region at different electric fields. a)** the PL spectra as a function of the filling factor at the zero electric field. **b** PL peak energy and **c** integrated PL intensity, extracted from (**a**), plotted as a function of the filling factor. **d–g** are doping-dependent PL spectra at several external electric fields increasing in the negative direction. **d, e** are in the quadrupolar exciton regime, while **f, g** are transitioned to the dipolar exciton regime. The dotted white lines in the color plots are the extracted PL peak energies through fitting (details in Supplementary Information Section 8). The PL data were taken from device D1.

filling factor denotes the number of holes per moiré unit cell ("−" sign for holes), the same as those in the moiré bilayer regions I and II. However, since the trilayer consists of two moiré superlattices, both of which can be filled with carriers, we define their individual filling factors as $n_t$ and $n_b$, respectively, and the total filling factor $n = n_t + n_b$. We focus on the low energy mode of the trilayer quadrupolar exciton and observe two main features in its PL spectra, at $n = -1$ and $n = -2$, respectively. At $n = -1$, the PL peak energy exhibits a kink (Fig. 3b), and the PL intensity drops sharply upon further hole doping (Fig. 3c). At $n = -2$, the PL energy exhibits a blueshift. These features correspond to the emergence of insulating states, similar to the previous studies[17,31–34].

The behaviors at these two fillings evolve systematically as a function of the external electric field. Since the device structure is symmetric, the observed PL behaviors are also symmetric with respect to the electric field direction. Figure 3d–g plot examples of PL spectra at selected negative electric fields (direction definition in Fig. 1a), while detailed data at both electric field directions are available in Supplementary Information Section 8. Note that the labeled values of external electric fields are calculated based on voltages applied on the top and bottom gates (see Methods). The effective electric fields between the top and bottom WSe₂ layers will be different due to carrier populations and layer chemical potentials (Supplementary Information Section 11).

As an electric field is applied, the PL spectra of the low energy quadrupolar mode remain largely unchanged concerning the two main features described above in the low field regime (Fig. 3d, e). However, it changes drastically at high electric fields (Fig. 3f, g): the PL intensity drops quickly when doped away from CNP, and the PL energy exhibits a blueshift at $n = -1$ instead of $n = -2$. In fact, the PL spectra at high electric fields resemble that of dipolar excitons in a moiré bilayer (Fig. 1e, f, as well as our previous study[26]). Therefore, the observed change in the PL spectra signals the transition from a quadrupolar exciton to a dipolar exciton. Similar results were reproduced in another device with the same structure (device D3), as shown in Supplementary Information Section 12.

With the understanding of the quadrupolar to dipolar exciton transition, we now discuss the nature of the $n = -1$ and $-2$ states and their evolution under electric fields. Figure 4a, c plot the PL intensity and peak energy as a function of both doping (filling factor) and external electric field, respectively. At $n = -1$, the PL energy and intensity both change abruptly above a certain threshold external electric field $E_{c,-1}$ (about 44 mV/nm), while at $n = -2$, the PL blueshift disappears when the external electric field exceeds $E_{c,-2}$ (about 32 mV/nm). For the $n = -1$ state, in the absence of an external electric field, each hole is hybridized between the top and bottom WSe₂ layers with

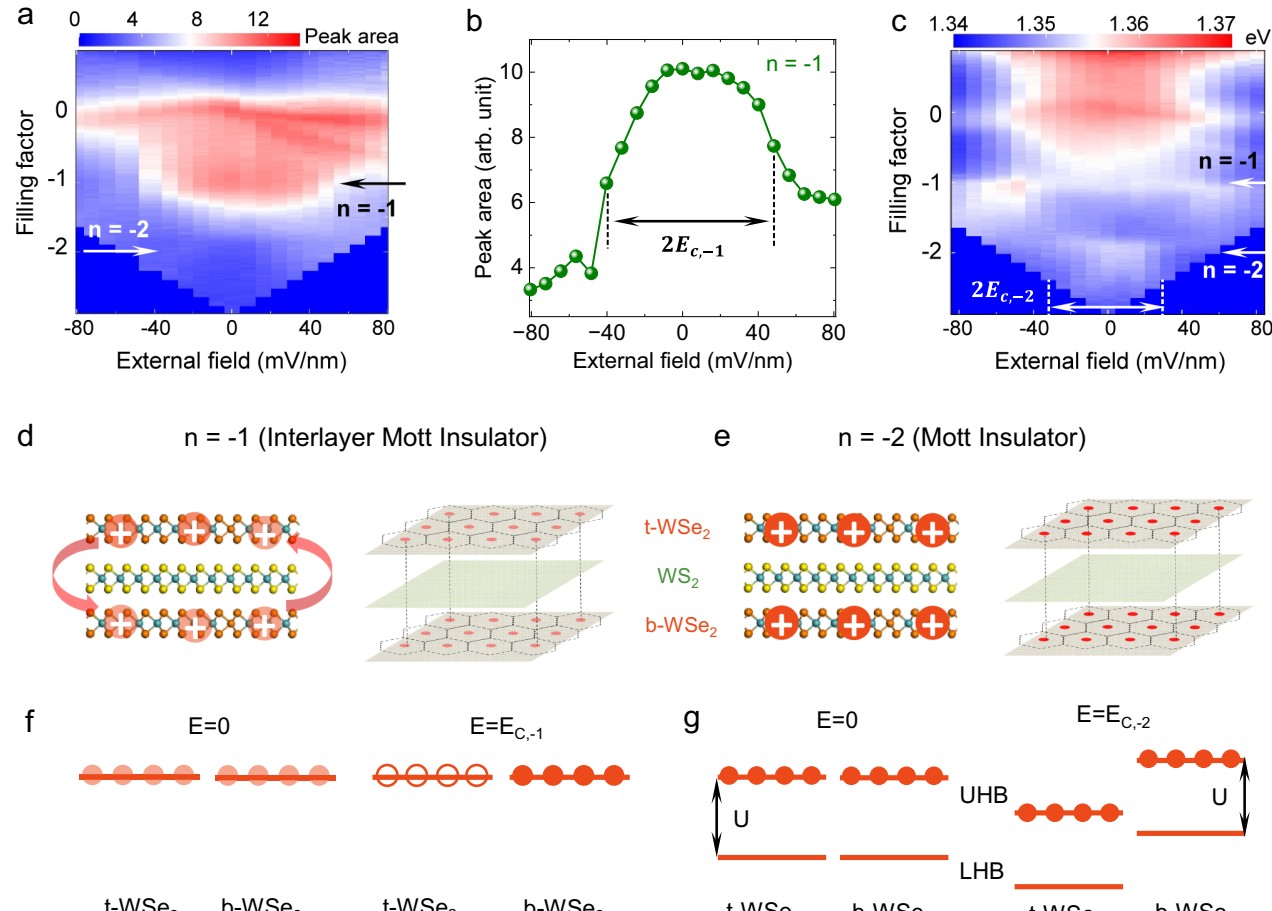

**Fig. 4 | Interlayer Mott insulator at $n = -1$ and Mott insulator at $n = -2$. a** The color plot of integrated PL intensity as a function of filling factor and external electric field. **b** The linecut of (**a**) at $n = -1$. **c** The color plot of PL peak energy as a function of the filling factor and external electric field. **d, e** are schematics of the hole configuration for the interlayer Mott insulator at $n = -1$ and Mott insulator at $n = -2$, respectively. **f, g** are the evolution of alignment and filling of the flat valence bands of the t-WSe$_2$ and b-WSe$_2$ layers as the electric field increases for $n = -1$ (**f**) and $n = -2$ (**g**). The PL data were taken from device D1.

equal probability, i.e., the hole wavefunction is a superposition of the top and bottom WSe$_2$ valence moiré bands. Laterally it is confined such that there is one hole in the two overlapping moiré unit cells combined (Fig. 4d). This state is a new type of correlated state in the trilayer moiré superlattice, an interlayer Mott insulator. The hole is allowed to tunnel between the top and bottom WSe$_2$ layers in the overlapping moiré cells, but tunneling to neighboring moiré cells is prohibited by the strong Coulomb repulsion. As the electric field increases, for example, in the positive direction defined in Fig. 1a, the probability of holes in the bottom WSe$_2$ layer will increase. Above the threshold electric field $E_{c,-1}$, the hole will be 100% in the bottom WSe$_2$ layer ($n_b = -1$), leaving the top WSe$_2$ layer empty ($n_t = 0$). This state now becomes a Mott insulator in the WS$_2$/b-WSe$_2$ interface only, similar to that in a WS$_2$/WSe$_2$ moiré bilayer. The system should remain insulating, as seen in the behavior of the PL peak energy in Fig. 4c. This transition is the result of the competition between the interlayer and intralayer hopping processes, which we characterize as energy $t'$ and $t$, respectively. The interlayer (intralayer) hopping favors carriers populating both (individual) WSe$_2$ layers. Based on the threshold electric field, we estimate the overall potential difference between the two WSe$_2$ layers is about 0 meV at the transition, which suggests that $t'$ is about the same as $t$ (See Supplementary Information Section 11: case 2 for a detailed discussion). We note that it is critical to have similar twist angles to observe the reported hybridized Mott insulator state here. The small difference in the twist angles of the reported device might lead to a moiré superlattice of a much larger period, which is not likely

to affect our experimental observation due to the corresponding low density of carriers for the half-filling.

At $n = -2$, the transition is different. Initially, at zero field, there is one hole per moiré unit cell in each of the two WSe$_2$ layers, forming two separate Mott insulator states at both t-WSe$_2$/WS$_2$ and WS$_2$/b-WSe$_2$ interfaces (Fig. 4e). Application of an electric field will create an energy shift between the two Mott insulator Hubbard bands. However, since both upper Hubbard bands (UHB) are fully occupied by holes, tunneling of holes between the two layers is forbidden, and this carrier configuration ($n_t = n_b = -1$) will remain stable until the UHB of the top WSe$_2$ layer starts to overlap with the lower Hubbard band (LHB) of the bottom WSe$_2$ layer (Fig. 4g), and holes from this top layer UHB will start to move to the LHB in the bottom WSe$_2$ layer, resulting in partially filled bands in both layers such that the system will no longer be insulating (see the $n = -2$ evolution in Fig. 4c). The energy difference between the two WSe$_2$ layer at the transition should be equal to the difference between the onsite Coulomb repulsion, U, and $t' - t$. This potential difference is estimated to be ~20 meV from the threshold field. As $t' - t$ is about 0, this suggests a value of about 20 meV for U, consistent with the previous studies[12,41]. We note that the threshold electric field at $n = -2$ has a large uncertainty due to the weak PL signals, and the resulting estimation of U is a lower bound.

Finally, the temperature-dependent PL spectra (Fig. S7) show that the interlayer Mott insulator transition temperature is about 80 K, consistent with our expectation based on previous studies on Mott

insulator state in WS$_2$/WSe$_2$ moiré systems[41,42]. The quadrupolar excitons, however, are still obvious at 100 K.

We note that we have also observed quadrupolar excitons and correlated states in WS$_2$/WSe$_2$/WS$_2$ trilayer moiré devices in which the conduction bands in the two WS$_2$ layers are hybridized (Supplementary Information Section 19). We choose to focus on the WSe$_2$/WS$_2$/WSe$_2$ trilayer system in this work as the hybridization and interlayer Mott insulator only involve one valence band in each WSe$_2$ monolayer instead of two conduction bands in each WS$_2$ monolayer, which simplifies the system.

In summary, our study demonstrates a unique trilayer moiré system that hosts both quadrupolar excitons and correlated states at $n = -1$ (interlayer Mott insulator) and $n = -2$ (Mott insulator). In particular, the quadrupolar excitons and interlayer Mott insulator both originate from the valence band hybridization and interact with each other. Here, the flat valence band hybridization, combined with the large spin-orbit coupling, is promising for generating nontrivial topological states and engineering quantum states such as quantum anomalous Hall[43]. The quadrupolar excitons in this unique trilayer moiré system are not only promising for realizing the quantum phase transition of bosonic quasiparticles but also strongly interact with correlated electrons, setting up an exciting platform for engineering new correlated physics of fermions, bosons, and a mixture of both[44]. We also envision that further development in aligning the moiré trilayer to allow different stacking of moiré sites (high symmetry points[45]) such as AAA, ABA, or ABC will usher in unprecedented opportunities in electronic and excitonic band engineering.

Note: During the submission of this work, we became aware of other works on quadrupolar excitons (ref. 46, ref. 47, and reference 29 in ref. 47).

## Method

### Sample fabrication

We used the same dry pick-up method as reported in our earlier work[32] to fabricate TMDC heterostructures. The gold electrodes are pre-patterned on the Si/SiO$_2$ substrate. The monolayer TMDC flakes, BN flakes, and few-layer graphene (FLG) flakes are exfoliated on silicon chips with 285 nm thermal oxide. It is worth noting that typical large TMDC flakes with one dimension exceeding 50 μm were chosen for the device structure shown in Fig. 1. The polycarbonate (PC)/ polydimethylsiloxane (PDMS) stamp was used to pick up TMDC monolayer and other flakes sequentially. The top WSe$_2$ and bottom WSe$_2$ are aligned with a 0-twist angle (R-stacked configuration). This is achieved either through angle-aligned layer stacking and checking the second harmonic generation (SHG) afterward or using the same WSe$_2$ flake and splitting it into two pieces via the tear and twist technique. The alignment of each layer is achieved under a home-built microscope transfer stage with the rotation controlled with an accuracy of 0.02 degrees. The PC is then removed in the chloroform/isopropanol sequence and dried with nitrogen gas. The final constructed devices were annealed in a vacuum (<10$^{-6}$ torr) at 250 °C for 8 h.

### Optical characterizations

During the optical measurements, the sample was kept in a cryogen-free optical cryostat (Montana Instruments). A home-built confocal imaging system was used to focus the laser onto the sample (with a beam spot diameter of ~2 μm) and collect the optical signal into a spectrometer (Princeton Instruments). During the measurements, the samples were kept in a vacuum and cooled down to 6–10 K. The PL measurements in Figs. 1, 2 are performed with 50 μW 633 nm CW excitation. All other PL measurements were performed with 633 nm CW excitation with a power of 200 μW unless specified. The reflectance contrast measurements were performed with a super-continuum laser (YSL Photonics). The polarized SHG measurements were performed with a pulsed laser excitation centered at 900 nm (Ti: Sapphire;

Coherent Chameleon) with a repetition rate of 80 MHz and a power of 80 mW. The angle between the laser polarization and the crystal axes of the sample was fixed. The SHG signal was analyzed using a half-waveplate and a polarizer. Additional PL measurements were performed with a 730 nm CW diode laser (Supplementary Information Section 15), which showed similar results as the main text.

### Calculation of electric field

The external electric field is defined as $\frac{1}{2}(\frac{V_{TG}}{d_1} - \frac{V_{BG}}{d_2})$, where $V_{TG}$ ($V_{BG}$) is the top (back) gate voltage, and $d_1$($d_2$) is the thickness of the top (bottom) layer BN flake.

The electric field in Fig. 2 is defined as the electric field in the TMDC heterostructure, which is given by $\frac{\varepsilon_{BN}}{2\varepsilon_{TMDC}}(\frac{V_{TG}}{d_1} - \frac{V_{BG}}{d_2})$.

## Data availability

Source data are available for this paper. The data in Figs. 1–4 are provided in the source data files. All other data that support the plots within this paper and other findings of this study are available from the corresponding author upon reasonable request. Source data are provided with this paper.

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

## Acknowledgements

We thank Prof. Chenhao Jin for the helpful discussions. Z.L. and S.-F.S. acknowledge support from NYSTAR through Focus Center-NY–RPI Contract C150117. The device fabrication was supported by the Micro and Nanofabrication Clean Room (MNCR) at Rensselaer Polytechnic Institute (RPI). S.-F.S. also acknowledges the support from NSF Grant DMR−1945420, DMR−2104902, and ECCS-2139692. X.H. and Y.-T.C. acknowledge support from NSF under awards DMR-2104805 and DMR-2145735. The optical spectroscopy measurements were supported by DURIP awards through Grant FA9550-20–1-0179 and FA9550-23-1-0084. S.T. acknowledges support from NSF DMR-1904716, DMR-1838443, CMMI-1933214, and DOE-SC0020653. K.W. and T.T. acknowledge support from JSPS KAKENHI (Grant Numbers 19H05790, 20H00354, and 21H05233). Y.S. and C.Z. acknowledge support from NSF PHY−2110212, PHY−1806227, OMR-2228725, ARO (W911NF17-1-0128), and AFOSR (FA9550−20–1-0220).

## Author contributions

S.-F.S and Z.L. conceived the project. Z.L., D.C., and Y.M. fabricated devices. Z.L., D.C., L.M., L.Y., X.H., and Q.W. performed measurements. M.B. and S.T. grew the TMDC crystals. T.T. and K.W. grew the BN crystals. S.-F.S., Y.-T.C., Z.L., D.C., X.C., Y.S., and C.Z. analyzed the data. S.-F.S and Y.-T.C. supervised the project. S.-F.S. and Y.-T.C. wrote the manuscript with inputs from all authors.

## Competing interests

The authors declare no competing interests.
