## [Peer Review File · Nature Communications]

Reviewers' Comments:

Reviewer #1:

Remarks to the Author:

The authors have conducted research on quadrupolar excitons in symmetric WSe₂/WS₂/WSe₂ trilayer structures. By applying z-electric fields, they have observed a quadratic dependence, which is different from the linear Stark shift observed in dipolar excitons. They have used electrostatic gating to adjust the electron and hole filling in the moire supercell, leading to the observation of interlayer Mott insulator states with carriers confined in moire sites. This work addresses a highly sought-after subject that is currently of great interest in the field of ordered interlayer exciton quantum phases. The authors have carried out numerous experiments and prepared multiple devices to ensure the consistency of their results.

However, I currently have some questions and concerns about the work and, therefore, cannot suggest it for publication at this stage. Furthermore, there are two highly relevant works published roughly half a year ago or more (<https://doi.org/10.48550/arXiv.2208.05490> and <https://doi.org/10.48550/arXiv.2207.09601>) that should also be taken into consideration. Despite this, I believe that the work of Z. Lian et al. deserves a careful review for possible publication in Nature Communications. Below, I outline my specific questions and concerns.

1. Is it possible that there is a quality issue with the heterostructure causing the PL shape as a function of doping to be different between areas I and II, and why are the energies of dipolar excitons so different between area I and area II? The difference in energy of 40 meV cannot be attributed to strain alone, so what other factors might be at play? Also, for the formation of quadrupolar excitons smaller energy mismatches have been considered in the supplementary material (section S10). What do the authors comment on this?
2. Can the authors clarify their confidence on the insulator states being $n=\pm 1, \pm 2$ and not any other fractional filling factors (or even generalized Wigner crystals)? Is it simply based on the conversion of the applied voltage to carrier density with a given thickness of the dielectric (hBN)?
3. Have the authors calculated what is the expected valence band splitting in the WSe₂/WS₂/WSe₂ structure? What is the predicted value of the splitting? Can they also perform a similar calculation of the Δ_{DQ} (as in reference 14) for this system, given that the deviation between 9 and 30 meV between different devices is significant? Can the authors explain this deviation?
4. Can the authors explain why the intensity of the dipolar exciton decreases for positive (area I) and negative (area II) values of the electric field?
5. It is noted that at large electric fields, the quadrupolar exciton approaches the linear Stark shift of dipolar excitons (page 3), but the explanation for this observation is unclear. Has the hybridization been modified as a function of the electric field, or is there a breakdown of the quadrupolar exciton into two dipolar excitons with opposite dipole moments? Also, why does the energy shift continue from the quadrupolar exciton state in this case?
6. It is unclear why the higher energy peak of area III cannot be a spectral signature of dipolar excitons shown in Figure 2a and 2b. Dipolar excitons should still form in the individual bilayers when the laser spot is on the trilayer, so it seems like there is an X-shape behavior in Figure 2c.
7. The reduced exciton-exciton repulsion of quadrupole excitons is not sufficiently explained to justify the smaller shifts as a function of the excitation power. Is this simply because of the smaller dipole moment (symmetric), and why is the lifetime similar to dipolar excitons (Figure S15)? One would expect shorter lifetimes for quadrupolar excitons because of the larger overlap in the electron-hole wavefunction.
8. The authors should briefly explain the microscopic mechanism of the quadrupolar exciton blueshift at $n=-1$ and $n=-2$ (Figures 3a and 3b) and why the shift is stronger for $n=-2$.
9. There is an assumption that the moire supercell of the trilayer will be similar to that of the individual bilayers, but the slight mismatch in the twist angle between the top and bottom bilayer may introduce a new periodicity in the trilayer that needs to be considered.
10. In page 5 it is stated: "Based on the threshold electric field, we estimate the overall potential difference between the two WSe₂ layers is about 0 meV at the transition, which suggests that t' is about the same as t (See SI for more discussion)." Where exactly is this discussion in the SI?
11. In page 6 it is mentioned: "We choose to focus on the WSe₂/WS₂/WSe₂ trilayer system in this work as the hybridization and interlayer Mott insulator only involve one valence band in each WSe₂ monolayer instead of two conduction bands in each WS₂ monolayer, which simplifies the

system.". Why is that? Due to smaller SO splitting in the conduction band? The simplicity is not clearly stated. In fact, in the reference <https://doi.org/10.48550/arXiv.2208.05490>, a trilayer WS₂/WSe₂/WS₂ system is examined. So, if a comparison is stated here it must be supported by a clear argument.

12. Have you tried to check reflectivity spectra close to the quadrupolar energies? Due to hybridisation of the valence states there might be some gain in the oscillator strength.

13. A general comment for maybe future experiments. It would be very interesting to measure the g-factor of the quadrupolar exciton. Since hybridization of the WSe₂ valence bands is included, I would expect a different value compared to dipolar excitons. Then, magnetic field dependent experiments as a function of electric field will demonstrate the transition from quadrupolar to dipolar excitons. Perhaps even information about the alignment between the layers will be extracted.

14. The manuscript is missing citations, especially in the section on exciton-correlated electron interactions where there are only self-citations, e.g. see Nature 587, 214–218 (2020). Other previous relevant work, such as that on MoS₂ homotrilayers with a quadrupole interlayer exciton, should be cited as well (see, Figure 4 of Nat. Nanotechnol. 15, 901–907 (2020)).

15. Typo in page 5: "Figs. 4a and 4c plot the PL peak intensity and energy as a function of..". Typo in the abstract "Transitional" and "valance". The authors should carefully proof read the text and supplementary for typos. For instance, in the supplementary Figure S9 legend, "phonon" instead of "photon".

Reviewer #2:

Remarks to the Author:

In this manuscript, the authors report the observation of quadrupolar exciton and interlayer Mott insulator in WSe₂-WS₂-WSe₂ trilayer moiré superlattice. The quadrupolar exciton has been theoretically predicted (Phys. Rev. Lett. 125, 255301, 2020) and experimentally observed (arXiv:2208.05490, 2022). And the hybridized interlayer correlated states also have been reported in WS₂/bilayer WSe₂/WS₂ heterostructures (Nature Materials, 22, 175–179, 2023), they even observed hybridized fractional correlated states. Therefore, in order to meet the high standard of Nature Communications, I could not recommend this manuscript in the current version. Below are my comments:

1. Page 3, the author said "as a result, the quadrupolar excitons will have two branches... Fig. 1g plots the PL in this region, which indeed exhibits two major PL resonances at energies below and above the dipolar excitons in Figs. 1e and 1f". When reading these sentences, the readers will think the two peaks separated by about 80 meV in Fig. 1g are from the hybridization of two dipolar excitons in Fig. 1b. However, when the authors fit the data of quadrupolar exciton in Fig. 2d, two branches are separated only by 9 meV, and the higher energy peak (at ~1.43 eV) has never been explained.

I understand that the authors fit the data only by using the lower energy quadrupolar exciton, then what is the higher energy peak (~1.43 eV)? In Fig. 2c, the higher energy peak also has quadrupolar exciton behavior, although it's not very clear.

2. Why there is no such a higher energy peak (~1.43 eV in Device 1) in other devices?

3. In device 1, the correlated states are clear in Region I, but not in Region II and III. This means the correlated states are spatial dependent, properly due to the inhomogeneity and strain. It's not convincing to use the correlated states in Region 1 to calibrate the filling factor in Region III, since there could be double moiré effect in Region III. Can the authors have a device to show clear correlated states in all three regions?

4. It's better to have same Y-axis scale for Figs. 1e, 1f, and 1g.

5. In Supplementary section 13, device D3 shows better correlated states. Can the authors compare the correlated states for all three regions in device D3?

6. Page 5, "Figs. 4a and 4c plot the PL peak energy and intensity as a function of both doping", to fit the data in Fig. 4, the sentence should be "Figs. 4a and 4c plot the PL intensity and peak energy as a function of both doping".

7. Page 4, "The integrated PL intensity of quadrupolar excitons exhibits more nonlinear dependence than dipolar excitons, likely due to their larger size". Can the authors explain this in more detail? And the power law fit is quite different from the results in Ref. 36, is there a reason?

8. The reference list should be improved. For example, Ref. 17 was published at the same time on

the same topic as the authors' paper Ref. 3, they should be cited together. Some of the references are in the wrong format, such as Ref. 29. And there are quite a lot of the references the author missed, like Xiaodong Xu's new paper (<https://www.nature.com/articles/s41563-023-01496-2>) and some other papers but not limited in this field: npj 2D Materials and Applications, 6, 79 (2022); Phys. Rev. Lett. 127, 037402, 2021; Nature Nanotechnology, 18, 233–237 (2023); Nature Materials, 22, 175–179 (2023); Nature Physics, 18, 395–400 (2022)

Reviewer #3:

Remarks to the Author:

The manuscript by Lian, Chen, et al. observes signatures of quadrupolar excitons in aligned trilayer stack of WSe₂/WS₂/WSe₂. The manuscript is well written and sufficient evidence is provided for the conclusions and claims. I recommend publication of this work if the following questions can be addressed.

1. When a WSe₂/WS₂ bilayer is stacked with 0-degree alignment, it exhibits a repeating pattern of AA, AB, BA stackings. However, with the addition of a third layer, numerous unique configurations can arise depending on the relative in-plane displacement of the third layer with respect to the first two. To determine these configurations, one can visually superimpose three meshes (two with the same lattice constant and one with a 4% mismatch). For instance, unique twisted trilayer graphene superlattices can be formed using high-symmetry AAA, ABA, or ABC stackings (arXiv:1907.12338). Nonetheless, the manuscript's findings may not be independent of the stacking pattern, and working with trilayers may require more samples to be prepared to account for the various possible configurations and ensure the generality of the observations. Future studies in this area would benefit from a description of the complexity of trilayer structures and starting point dependence.

2. The interlayer and intralayer excitons in the bilayer WSe₂/WS₂ heterostructure are shown to be localized (Nature 603, 247–252 (2022), Nature 609, 52–57 (2022), Science 378, 1235–1239 (2022)). Are the excitons in the trilayer expected to be localized as well? What is the influence of the moiré potential and reconstruction on the quadrupolar exciton?

3. Why do Fig. 1 e and f, which are both for dipolar excitons, show quite different doping dependences?

4. The author should comment on how the purple spheres are extracted in Fig. 2(d).

We sincerely thank the reviewers for their time and efforts, and we greatly appreciate the reviewers' recognition of our work and their constructive comments. Below we provide a point-to-point reply to reviewers' comments, and we have significantly revised our manuscript accordingly. With the revisions, we believe that we have fully addressed the reviewers' questions and our manuscript is ready for publication in Nature Communications. We thank all the reviewers for helping us improve our manuscript.

REVIEWER COMMENTS

Reviewer #1 (Remarks to the Author):

The authors have conducted research on quadrupolar excitons in symmetric WSe₂/WS₂/WSe₂ trilayer structures. By applying z-electric fields, they have observed a quadratic dependence, which is different from the linear Stark shift observed in dipolar excitons. They have used electrostatic gating to adjust the electron and hole filling in the moire supercell, leading to the observation of interlayer Mott insulator states with carriers confined in moire sites. This work addresses a highly sought-after subject that is currently of great interest in the field of ordered interlayer exciton quantum phases. The authors have carried out numerous experiments and prepared multiple devices to ensure the consistency of their results.

However, I currently have some questions and concerns about the work and, therefore, cannot suggest it for publication at this stage. Furthermore, there are two highly relevant works published roughly half a year ago or more (<https://doi.org/10.48550/arXiv.2208.05490> and <https://doi.org/10.48550/arXiv.2207.09601>) that should also be taken into consideration. Despite this, I believe that the work of Z. Lian et al. deserves a careful review for possible publication in Nature Communications. Below, I outline my specific questions and concerns.

We thank the reviewer for the recognition of our work. We also want to bring to the reviewer's attention that our work was initially submitted around the same time, or even earlier than the arxiv paper the reviewer mentioned (<https://doi.org/10.48550/arXiv.2208.05490>). One of the earlier comments from a reviewer mentioned the following:

"There are similar papers that appeared on arXiv recently (arXiv:2208.05490v1 and its ref 29). The authors might consider mentioning them. However, I acknowledge that these were posted after Z. Lian et al.,'s submission to Nature Physics."

We also bring to the reviewer the attention that the mentioned arxiv paper mainly studies the electric field dependence of PL spectra to confirm the existence of the quadrupolar excitons. Our work not only studies the electric field dependence on the PL spectra but also investigates the doping dependence, which reveals the new hybridized Mott insulator state.

In the revised manuscript, we include the citation of <https://doi.org/10.48550/arXiv.2207.09601> as

"Note: During the submission of this work, we became aware of other works on quadrupolar excitons (Ref.⁴⁵, Ref.⁴⁶ and its reference 29)."

1. Is it possible that there is a quality issue with the heterostructure causing the PL shape as a function of doping to be different between areas I and II, and why are the energies of dipolar excitons so different between area I and area II? The difference in energy of 40 meV cannot be attributed to strain alone, so what other factors might be at play? Also, for the formation of quadrupolar excitons smaller energy mismatches have been considered in the supplementary material (section S10). What do the authors comment on this?

We first address the most important question: the formation of quadrupolar excitons with possible energy mismatch of the dipolar excitons. Typically we need a large sample size to ensure three distinct regions. The sample shown in the original Fig. 1 is more than 100 μm in one direction, much larger than most TMDC moiré devices! For example, the sample in the mentioned arxiv paper only shows one dipolar exciton region. Due to this large size and possible spatial inhomogeneity, further complicated by the different dielectric environment, the energy of the dipolar excitons involved in the quadrupolar excitons formation could be different from that from Region I and Region II.

However, the existence of quadrupolar exciton (characterized by the quadratic rather than linear electric field dependence) suggests similar dipolar exciton energies. Further, using the analysis in SI Section S10, which we also show here as Fig. R1 for the reviewer's convenience, a slight difference in dipolar exciton energy can be found in the E field dependence of the quadrupolar excitons. If the dipolar exciton energies are the same, the hybridization could occur at zero electric fields. If the dipolar exciton energies are different, we need a finite electric field to bring the exciton energies to be the same to ensure optimum hybridization (Fig. R1). As a result, the electric field dependence of the quadrupolar exciton can directly extract the dipolar excitons' energy difference at the spot where the quadrupolar excitons are formed (Region III), independent of the measurement results from Region I and II (details in SI Section S10). All quadrupolar exciton devices we measured exhibit energy differences of less than 7 meV, with most less than 3 meV.

Figure R1. A schematic showing the hybridization of dipolar excitons with different energies at zero electric field.

With the above discussion, we agree that the dipolar excitons energy difference between the region I and II is large (but within the statistics that we have observed over many different samples, shown in SI). As the moiré potentially is largely determined by the atomic reconstruction [Nat. Mater. 20, 945–950 (2021)], we suspect that the strain plays an important role. We are not clear on other possible reasons for the energy difference yet but will explore that in the future.

Here we also present the data from a new device with reduced dipolar exciton energy differences between Region I and Region II. The electric field dependence of the quadrupolar excitons (shown in Fig. R2) also shows that the two dipolar excitons have similar energies. We have thus replaced our original Fig. 1 in the manuscript with Fig. R2. We have also performed electric field dependence on the same device (Device D5) here, shown in Fig. R3. We have replaced Fig.2 of the main manuscript with Fig. R3.

Figure R2. Excitons in different stacking structures of the trilayer device. (a) Schematics of the device structure with three different regions: (I) WS₂/b-WSe₂ (II) t-WSe₂/WS₂/b-WSe₂ (III) top bilayer, t-WSe₂/WS₂. (b) Schematics of the dipolar and quadrupolar excitons configuration. (c) Type-II alignment of the angle-aligned WSe₂/WS₂ heterobilayer. (d) Valence band hybridization in the trilayer region, compared with the flat valence band of WSe₂ in the WSe₂/WS₂ moiré bilayer regions. (e), (f), and (g) are doping-dependent PL spectra for regions I, II, and III.

Figure R3. Electric field dependent PL spectra of dipolar and quadrupolar excitons. (a), (b), and (c) are electric field dependent PL spectra of the region I, II, and III, respectively. (d) Fitting of the quadrupolar excitons PL resonances (orange curve) on extracted PL peak energy (purple spheres) as a function of the electric field from (c).

2. Can the authors clarify their confidence on the insulator states being $n=\pm 1, \pm 2$ and not any other fractional filling factors (or even generalized Wigner crystals)? Is it simply based on the conversion of the applied voltage to carrier density with a given thickness of the dielectric (hBN)?

The assignment of the correlated states is not from the capacitive model calculation. The insulating states can be reliably read out from the doping-dependent reflectance spectroscopy study in terms of intensity change, shown in Fig. R3 (device D1). They can also be read out by the doping-dependent PL spectra as the intensity changes and peak shifts at these insulating states (Fig. R3). In the previous studies, we have also systematically established the correlation between the microwave impedance microscopy measurements and doping-dependent

reflectance (PL) spectra to help us confidentially assign these incompressible states in the bilayer moiré superlattice using doping-dependent PL spectra (Nat Commun 12, 3608 (2021)). These assignments are also found to be consistent with the calculated filling factor based on the capacitance model.

Figure R4. Doping-Dependent Reflection (a) and PL (b) spectra from region I of device D1.

3. Have the authors calculated what is the expected valence band splitting in the $WSe_2/WS_2/WSe_2$ structure? What is the predicted value of the splitting? Can they also perform a similar calculation of the Δ_{DQ} (as in reference 14) for this system, given that the deviation between 9 and 30 meV between different devices is significant? Can the authors explain this deviation?

Without considering the moiré effect, we would not expect the results to be different from the calculation in reference 14, which predicts the Δ_{DQ} to be 10-30 meV. Our results are consistent with the calculations in reference 14.

The difference might arise from how close these dipolar excitons are coupled, which should be sensitive to the twist angle and the physical distance between the two layers (including the atomic reconstruction details). With all these factors considered, we would expect variations of Δ_{DQ} from device to device.

Reference 14 did not consider the moiré effect, and it will be interesting to investigate that theoretically, which is out of the scope of this work. We hope that our results could be a guide for future endeavors.

4. Can the authors explain why the intensity of the dipolar exciton decreases for positive (area I) and negative (area II) values of the electric field?

The mentioned experimental observations are expected considering the electric field's effect on the electron and hole recombination. The positive electric field in area I or negative electric field in area II (direction defined in the main text) will effectively separate the electron and hole

constituting the dipolar interlayer exciton (schematically shown below in Fig. R5), which will decrease the recombination rate and thus reduces the interlayer exciton PL intensity.

Figure R5. Electric field tuning of dipolar interlayer exciton recombination in the region I and II. (a), (b) and (c) show the dipolar interlayer excitons in region I under positive, zero, and negative electric field, respectively. A positive electric field drives electrons and holes away, inhibiting the recombination process, while a negative electric field drives electrons and holes toward each other, facilitating the recombination process. (d), (e), (f) show the dipolar interlayer excitons in region II under positive, zero, and negative electric field, respectively. A positive electric field facilitates recombination while a negative electric field inhibits recombination in region II. The above picture is consistent with the observed PL intensity change under the electric field in Region I and Region II.

5. It is noted that at large electric fields, the quadrupolar exciton approaches the linear Stark shift of dipolar excitons (page 3), but the explanation for this observation is unclear. Has the hybridization been modified as a function of the electric field, or is there a breakdown of the quadrupolar exciton into two dipolar excitons with opposite dipole moments? Also, why does the energy shift continue from the quadrupolar exciton state in this case?

The hybridization depends on the energy difference between the two dipolar interlayer excitons with opposite parity, which can be tuned by the electric field. One can intuitively imagine that for a small electric field, likely characterized by $E \ll \Delta/d$, where Δ is the coupling strength, the hybridization of the two dipolar excitons forms the quadrupolar excitons. For $E \gg \Delta/d$, the hybridization is negligible, and the quadrupolar excitons break down to two dipolar excitons. In between, the valence band hybridization is expected to be asymmetric, with hole wave function distribution in the top and bottom WSe_2 layer not equal.

The quadrupolar exciton energy remains a function of the electric field, being a quadratic function of the electric field in the symmetric hybridization case, while the finite dipolar moment contribution will kick in when the hybridization starts to be asymmetric.

The quadrupolar excitonic state can be expressed as the linear combination of the two dipolar excitonic states with opposite dipole moments, as described by the two-level hybridization model in Supplementary Section 6. At small electric fields, the quadrupolar exciton is expected to have an equal composition of the two dipolar excitons, whereas it approaches 100% dipolar exciton composition at the high electric field limit. Between these two extremes, the composition of the quadrupolar exciton is continuously tunable by the electric field.

6. It is unclear why the higher energy peak of area III cannot be a spectral signature of dipolar excitons shown in Figure 2a and 2b. Dipolar excitons should still form in the individual bilayers when the laser spot is on the trilayer, so it seems like there is an X-shape behavior in Figure 2c.

We thank the reviewer for the insightful suggestion. We do not know the exact nature of the high energy peak in Region III yet but believe that they are not the dipolar excitons shown in Region I and II as they have different dependencies on the electric field.

We have replaced the Figs.1 and 2 in the original manuscript with data from another device (device D5), which does not show the high energy PL in Region III under excitation power 50 μW using 1.96 eV CW excitation.

The nature of the high-energy peak is interesting but beyond the scope of this work. We will explore it in the future.

7. The reduced exciton-exciton repulsion of quadrupole excitons is not sufficiently explained to justify the smaller shifts as a function of the excitation power. Is this simply because of the smaller dipole moment (symmetric), and why is the lifetime similar to dipolar excitons (Figure S15)? One would expect shorter lifetimes for quadrupolar excitons because of the larger overlap in the electron-hole wavefunction.

The reviewer is correct in noting that the smaller dipole moment can explain the smaller shift as a function of the excitation power, while we believe the smaller dipole moment of the quadrupolar excitons also contributes to the reduced exciton-exciton repulsion, considering the original dipole of the excitons is fixed in polarity due to the electron-hole separation in two different layers.

The lifetime of the quadrupolar excitons in Region III in Fig. S15 is indeed smaller than that of dipolar excitons in Region II. We suspect that the reviewer meant that the difference was smaller than what he/she expected. The reviewer is correct in noting that the recombination rate is supposed to increase for the radiative recombination of quadrupolar excitons. However, our time-resolved PL measures the total lifetime that includes the contribution from both radiative and nonradiative channels. In addition, spatial inhomogeneity might also come into play in terms of lifetime. As a result, we believe that the extracted lifetime only serves as a check for the quadrupolar exciton, and our results are consistent with the expectation.

8. The authors should briefly explain the microscopic mechanism of the quadrupolar exciton blueshift at $n=-1$ and $n=-2$ (Figures 3a and 3b) and why the shift is stronger for $n=-2$.

We thank the reviewer for the question. As we discussed in the manuscript, the $n=-2$ state corresponds to the case that both top and bottom moiré bilayers are in the Mott insulator states,

with each of the WSe_2 layers half filled. The $n=-1$ state is the newly discovered hybridized Mott insulator state, in which the top and bottom WSe_2 share holes, while the hole maintains such density that that one superlattice (regardless of the layers) is occupied by one hole (about $2 \times 10^{12} \text{ cm}^{-2}$).

We suspect that the difference in the PL peak shift is related to the nature of the different correlated states, which leads to different band gaps.

The $n=-2$ state in Fig. 3 corresponds to the case where the holes in the top and bottom WSe_2 have half filling in each moiré superlattice (-1 for the top WSe_2/WS_2 and -1 for the bottom WS_2/WSe_2 moiré superlattice). Therefore, the bandgap will be the smaller onsite repulsion energy (U) of the top and bottom Mott insulator minus the energy Δ^\pm . In Fig. R6d, we assume the same U for both valence bands for simplicity. We believe that this U could be more than 50 meV (from one of our manuscripts under preparation and recent work on arXiv: arXiv:2209.12830v2).

The $n=-1$ state in Fig. 3 is the newly discovered hybridized Mott insulator state. It can be understood schematically as shown in Fig. R6. The hybridization of the flat valence band gives to the bonding state (share the origin of the quadrupolar excitons) and antibonding state, separated by the energy gap Δ^\pm . For the $n=-1$ state, the bonding state will be half filled as the LHB, while the UHB of the bonding state will be at the energy U higher. As we extracted Δ to be 9 meV (SI Section 10) for device D1, which corresponds to a Δ^\pm of 18 meV, the UHB will be higher than the antibonding state, so that the next added electron will occupy the antibonding state, leading to the bandgap of Δ^\pm . As Δ^\pm is smaller than $U - \Delta^\pm$, we would expect the shift at $n=-1$ to be smaller than that at the $n=-2$.

We have included this discussion in the revised SI. A more quantitative understanding of the peak shift would need to consider the possible exciton binding energy change, and we will explore that in the future.

Figure R6. Schematics showing the interlayer hybridized Mott insulating state. (a) shows the valence band hybridization process in $\text{WSe}_2/\text{WS}_2/\text{WSe}_2$ trilayers. (b), (c) and (d) show the schematics of the bandstructure in the trilayer region at hole filling of $n=0$, $n=-1$ and $n=-2$, respectively.

9. There is an assumption that the moiré supercell of the trilayer will be similar to that of the individual bilayers, but the slight mismatch in the twist angle between the top and bottom bilayer may introduce a new periodicity in the trilayer that needs to be considered.

The reviewer is correct in that the twist angle difference between the top bilayer and bottom bilayer can potentially generate another moiré superlattice of much larger periodicity.

Currently, we found that the correlated hybridized Mott insulator state occurs in fewer devices than the quadrupolar excitons states, and we suspect that the hybridized Mott insulator state is more sensitive to the twist angle difference. For device D1 we show in the original manuscript, the twist angle of the top and bottom two layers is 0.3° . The small twist angle difference would lead to a superlattice period of more than one order of magnitude than the ~ 8 nm moiré superlattice. The half-filling of this superlattice will require a density of carriers of about orders of magnitude smaller, which is much less than unintentional doping.

We have added the following statement to the manuscript:

“We note that it is critical to have a similar twist angle between the top and bottom layers to observe the reported hybridized Mott insulator state here. The small difference of the twist angle of the reported device might lead to a moiré superlattice of much large period, which is unlikely to affect our experimental observation due to the corresponding low moiré density (one order of magnitude smaller than individual moiré superlattice).”

10. In page 5 it is stated: “Based on the threshold electric field, we estimate the overall potential difference between the two WSe_2 layers is about 0 meV at the transition, which suggests that t' is about the same as t (See SI for more discussion).” Where exactly is this discussion in the SI?

The related discussion is included in the SI Section 8, case 2.

We have revised our manuscript as

“(See Supplementary Information Section 8: case 2 for a detailed discussion)”

11. In page 6 it is mentioned: “We choose to focus on the $\text{WSe}_2/\text{WS}_2/\text{WSe}_2$ trilayer system in this work as the hybridization and interlayer Mott insulator only involve one valence band in each WSe_2 monolayer instead of two conduction bands in each WS_2 monolayer, which simplifies the system.”. Why is that? Due to smaller SO splitting in the conduction band? The simplicity is not clearly stated. In fact, in the reference <https://doi.org/10.48550/arXiv.2208.05490>, a trilayer $\text{WS}_2/\text{WSe}_2/\text{WS}_2$ system is examined. So, if a comparison is stated here it must be supported by a clear argument.

In WSe_2 and WS_2 monolayer, the spin-orbit coupling induced conduction band splitting leads to two conduction bands close in energy, 20-40 meV. In contrast, the valence band splitting leads to valence band separation of 300-500 meV, so we only need to consider one valence band (Phys. Rev. Lett. 121, 026402). For the consideration of the discussion of correlated states, we believe it is simpler to focus on $\text{WSe}_2/\text{WS}_2/\text{WSe}_2$ trilayer system.

We are not surprised that the $\text{WS}_2/\text{WSe}_2/\text{WS}_2$ also exhibits the formation of quadrupolar excitons, as we have also observed it in $\text{WS}_2/\text{WSe}_2/\text{WS}_2$ trilayers. However, the interlayer hybridized Mott insulator state will involve electron bands in the two WS_2 layers which are more complicated than

the hole bands. We thus choose to focus on the $\text{WSe}_2/\text{WS}_2/\text{WSe}_2$ structure first. We will explore $\text{WS}_2/\text{WSe}_2/\text{WS}_2$ trilayers in detail in the future.

12. Have you tried to check reflectivity spectra close to the quadrupolar energies? Due to hybridisation of the valence states there might be some gain in the oscillator strength.

We have measured the reflectance contrast spectra near the resonance of quadrupolar exciton, and we do not observe any noticeable features.

Figure R7. Gate-dependent reflectance contrast spectra near the quadrupolar exciton resonance from the region III of device D5.

In fact, we would not expect much-enhanced oscillator strength from the hybridization of the two dipolar excitons. The top and bottom interlayer excitons dipolar are both much smaller in oscillator strength than that of intralayer excitons (more than 100 times smaller, *Science***376**,406-410(2022), *Nano Lett.* 2017, 17, 2, 638–643). If there is any increase of oscillator strength due to the hybridization, it will be less than twice of the original interlayer excitons oscillator strength considering the conservation of oscillator strength and the antisymmetric quadrupolar exciton state (related to the antibonding state), which is still a very small number and hard to probe directly from reflectance spectra.

13. A general comment for maybe future experiments. It would be very interesting to measure the g-factor of the quadrupolar exciton. Since hybridization of the WSe_2 valence bands is included, I would expect a different value compared to dipolar excitons. Then, magnetic field dependent experiments as a function of electric field will demonstrate the transition from quadrupolar to dipolar excitons. Perhaps even information about the alignment between the layers will be extracted.

We thank the reviewer for the suggestion. In a noninteracting picture, we actually expect the quadrupolar excitons to have a similar Zeeman shift as dipolar exciton in the low magnetic field regime (up to ~ 10 T), as we can think of the Zeeman shift as just the difference between the valence band shift minimum the conduction band shift in the magnetic field. As the two valences bands (the VBM of top and bottom WSe_2) involved in the hybridization that leads to the formation

of quadrupolar excitons should have the same Zeeman shift (same valley, orbital and spin contributions), we do not expect the Zeeman shift to be much different.

However, this picture does not include possible interactions that could change the above discussion, which will be intriguing to study. Also, we suspect the diamagnetic shift, which will be significant at high B field due to the quadratic B field dependence, will be different for quadrupolar excitons and dipolar excitons due to their different sizes.

These intriguing experiments are clearly out of the scope of the current work, but we will explore them in the future, and we thank the reviewer for the suggestion.

14. The manuscript is missing citations, especially in the section on exciton-correlated electron interactions where there are only self-citations, e.g. see Nature 587, 214–218 (2020). Other previous relevant work, such as that on MoS₂ homotrilayers with a quadrupole interlayer exciton, should be cited as well (see, Figure 4 of Nat. Nanotechnol. 15, 901–907 (2020)).

We thank the reviewer for the suggestions. We have cited the mentioned references in the revised manuscript(Ref. 11 and 39).

15. Typo in page 5: “Figs. 4a and 4c plot the PL peak intensity and energy as a function of..”. Typo in the abstract “Transitional” and “valance”. The authors should carefully proof read the text and supplementary for typos. For instance, in the supplementary Figure S9 legend, “phonon” instead of “photon”.

We sincerely thank the reviewer for careful reading of our manuscript. We have corrected the typos mentioned, and we have also proofread the paper carefully and corrected typos/grammar mistakes.

Reviewer #2 (Remarks to the Author):

In this manuscript, the authors report the observation of quadrupolar exciton and interlayer Mott insulator in WSe_2 - WS_2 - WSe_2 trilayer moiré superlattice. The quadrupolar exciton has been theoretically predicted (Phys. Rev. Lett. 125, 255301, 2020) and experimentally observed (arXiv:2208.05490, 2022). And the hybridized interlayer correlated states also have been reported in WS_2 /bilayer WSe_2/WS_2 heterostructures (Nature Materials, 22, 175–179, 2023), they even observed hybridized fractional correlated states. Therefore, in order to meet the high standard of Nature Communications, I could not recommend this manuscript in the current version. Below are my comments:

We want to bring to the reviewer's attention that the mentioned reference [Phys. Rev. Lett. 125, 255301, 2020] did not consider the moiré coupling, so it is not clear how the quadrupolar exciton will be formed or not. Also, it cannot predict correlated states and the interaction with the quadrupolar excitons, as no moiré flat band is involved in the theory.

We also want to bring to the reviewer's attention that our manuscript was initially submitted around the same time, or even earlier than the arxiv paper (arXiv:2208.05490, 2022, please see our reply to reviewer 1). It is just that we did not post it on arxiv. Our work not only demonstrates the quadrupolar exciton through electric field-dependent PL spectra but also includes the doping dependence study of the quadrupolar excitons that leads to the discovery of the hybridized Mott insulator state, which is not included in the mentioned arxiv paper [arXiv:2208.05490, 2022].

The hybridized Interlayer Mott state is here is completely different from the mentioned WS_2 /bilayer WSe_2/WS_2 heterostructures [Nature Materials, 22, 175–179, 2023]. The observed integer correlated state in the mentioned reference is an excitonic insulator, similar to previous works including one from us [Nat. Phys. 18, 395–400 (2022), Nat. Phys. 18, 1171–1176 (2022), Nat. Phys. 18, 1214–1220 (2022)], while what we report is a hybridized Mott insulator state. In fact, the distinct physics involved in our work compared to the Nature Materials paper requires a completely opposite design of experiment. The mentioned work requires the top WS_2 and bottom WS_2 layer to be a 60-degree twist to suppress interlayer tunneling regardless of the electric field, therefore, at the interlayer correlated states, occupied sites in the top WS_2 layer are correlated with the empty sites in the bottom WS_2 layer, in other words, each individual electron has 0 or 100% probability in either top or bottom layer and there is no "hybridization" between the two layers. In our case, the top and bottom WSe_2 layers need to be 0-degree aligned to allow maximum hole tunneling between the top and bottom WSe_2 layers, and each individual hole is hybridized between the top and bottom layers, and their wavefunction is split between the two layers. For example, at zero electric field, the probability of finding each hole in the top layer is 50%, and 50% in the bottom layer.

1. Page 3, the author said "as a result, the quadrupolar excitons will have two branches... Fig. 1g plots the PL in this region, which indeed exhibits two major PL resonances at energies below and above the dipolar excitons in Figs. 1e and 1f". When reading these sentences, the readers will think the two peaks separated by about 80 meV in Fig. 1g are from the hybridization of two dipolar excitons in Fig. 1b. However, when the authors fit the data of quadrupolar exciton in Fig. 2d, two branches are separated only by 9 meV, and the higher energy peak (at ~1.43 eV) has never been explained.

I understand that the authors fit the data only by using the lower energy quadrupolar exciton, then what is the higher energy peak (~ 1.43 eV)? In Fig. 2c, the higher energy peak also has quadrupolar exciton behavior, although it's not very clear.

We do not fully understand the nature of the high energy peak in the original Fig. 1, and it is possible that it is a quadrupolar exciton mode from higher energy dipolar excitons. Since the high energy peak is not universal across all samples and the new Fig.1 does not show that, we leave the discussion in the revised SI and will explore it in the future.

2. Why there is no such a higher energy peak (~ 1.43 eV in Device 1) in other devices?

As we mentioned above, high-energy PL occurs at high excitation power and is not universal among all the devices we have studied. It could be attributed to other exciton modes at higher energy, including possible quadrupolar exciton modes from hybridization of higher energy dipolar excitons. We do not believe that it is from the high energy branch (antisymmetric mode associated with the symmetric quadrupolar exciton mode discussed in the main text) quadrupolar exciton because of its large energy separation from the ground state quadrupolar exciton. We now include the discussion in the revised SI.

3. In device 1, the correlated states are clear in Region I, but not in Region II and III. This means the correlated states are spatial dependent, properly due to the inhomogeneity and strain. It's not convincing to use the correlated states in Region 1 to calibrate the filling factor in Region III, since there could be double moiré effect in Region III. Can the authors have a device to show clear correlated states in all three regions?

We have replaced the data Figure 1 with the that from another (D5), as shown in Fig. R2 (our reply to reviewer 1). We believe the correlated states are clear in all three regions now. Correlated states in all three regions can also be found in device D3, as shown in Fig. R9, in response to the reviewer's comment 5.

We also want to mention that the original Fig.1 show correlated states in region III clearly, without the need to use the correlated states in region I as the calibration. Here we show the gate dependence of the PL peak position and peak intensity, which is also shown in the Fig.3 of the main text. In Fig. R8, we compare the extracted PL intensity and the peak position with reflectance spectra measured from region I of device D1. The jumps of the PL peak position and PL intensity

can be clearly seen at integer fillings, in excellent agreement with independent reflectance spectra from Region I.

Figure R8. Comparison of the PL peak position (a) and the peak intensity (b) measured from region III of device D1 and the reflectance spectra (c) from region I of device D1. (a) and (b) are the data from Fig.3 of the main text and the data in (c) is from Fig.S3.

4. It's better to have same Y-axis scale for Figs. 1e, 1f, and 1g.

We thank the reviewer for the suggestion, and we have fixed it in the new Fig. 1

5. In Supplementary section 13, device D3 shows better correlated states. Can the authors compare the correlated states for all three regions in device D3?

We show in Fig. R9 the doping dependence of PL spectra from region I, II and III of device D3. It can be found that in Region I and II, the PL peaks of dipolar excitons exhibit blueshifts and intensity changes at filling factors corresponding to $n = 1$, -1 and -2 . The PL peak in region III exhibits an intensity change at the gate voltage corresponding to $n = -1$, and blueshift at the gate voltage corresponding to $n = -2$. These observations are consistent with the data we showed in Fig. 3 of the main text.

Figure R9. Comparison of the doping dependence of PL spectra from regions I, II, and III of device D3. The PL spectra were taken using $10\ \mu\text{W}$ $1.96\ \text{eV}$ CW excitation. The temperature is $5\ \text{K}$.

6. Page 5, “Figs. 4a and 4c plot the PL peak energy and intensity as a function of both doping”, to fit the date in Fig. 4, the sentence should be “Figs. 4a and 4c plot the PL intensity and peak energy as a function of both doping”.

We thank the reviewer for the careful reading and we have revised our manuscript accordingly.

7. Page 4, “The integrated PL intensity of quadrupolar excitons exhibits more nonlinear dependence than dipolar excitons, likely due to their larger size”. Can the authors explain this in more detail? And the power law fit is quite different from the results in Ref. 36, is there a reason?

We expect the quadrupolar excitons to be larger in size than the dipolar excitons, considering the sharing of the holes between the top and bottom layers. As a result, we expect the excitation power to reach one exciton per exciton radius is lower for the trilayer region. As a result, nonlinear effects such as absorption saturation or an increase in nonradiative channels might occur for the trilayer region, which leads to the more nonlinear power dependence of the PL of quadrupolar excitons.

The ref. 36 has different trilayer structures, $\text{WS}_2/\text{WSe}_2/\text{WS}_2$, and the quadrupolar excitons are formed through the hybridization of the conduction band and might be different from ours. It is also not clear why the ref. 36 did the power dependence study in a very high power regime, from $0.5\ \text{mW}$ to $4\ \text{mW}$. Under such high excitation power, the quadrupolar exciton is supposed to break down and becomes dipolar excitons (possibly forming staggering dipolar excitons lattice). It is not clear whether the power-dependent spectra are still from quadrupolar excitons.

8. The reference list should be improved. For example, Ref. 17 was published at the same time on the same topic as the authors’ paper Ref. 3, they should be cited together. Some of the references are in the wrong format, such as Ref. 29. And there are quite a lot of the references the author missed, like Xiaodong Xu’s new paper (<https://www.nature.com/articles/s41563-023-01496-2>) and some other papers but not limited in this field: npj 2D Materials and Applications, 6, 79 (2022); Phys. Rev. Lett. 127, 037402, 2021; Nature Nanotechnology, 18, 233–237 (2023); Nature Materials, 22, 175–179 (2023); Nature Physics, 18, 395–400 (2022)

We thank the reviewer for the suggestion. We have revised our original reference and added the new references. We thank the reviewer for helping us improve our manuscript.

Reviewer #3 (Remarks to the Author):

The manuscript by Lian, Chen, et al. observes signatures of quadrupolar excitons in aligned trilayer stack of $WSe_2/WS_2/WSe_2$. The manuscript is well written and sufficient evidence is provided for the conclusions and claims. I recommend publication of this work if the following questions can be addressed.

We thank the reviewer for the recognition of our work.

1. When a WSe_2/WS_2 bilayer is stacked with 0-degree alignment, it exhibits a repeating pattern of AA, AB, BA stackings. However, with the addition of a third layer, numerous unique configurations can arise depending on the relative in-plane displacement of the third layer with respect to the first two. To determine these configurations, one can visually superimpose three meshes (two with the same lattice constant and one with a 4% mismatch). For instance, unique twisted trilayer graphene superlattices can be formed using high-symmetry AAA, ABA, or ABC stackings (arXiv:1907.12338). Nonetheless, the manuscript's findings may not be independent of the stacking pattern, and working with trilayers may require more samples to be prepared to account for the various possible configurations and ensure the generality of the observations. Future studies in this area would benefit from a description of the complexity of trilayer structures and starting point dependence.

We agree with the reviewer that a trilayer structure can exhibit a large variety of lattice stacking configurations. We have observed quadrupolar excitons in more than 16 samples while the interlayer hybridized Mott insulator state only appears in a fraction of them. We thus suspect that the formation of quadrupolar exciton is less susceptible to trilayer stacking since the moiré superlattice is not a necessary requirement, while the hybridized Mott insulator will certainly be sensitive to the layer stacking. We appreciate the reviewer's suggestion and will investigate along this direction in more detail in our future work.

The trilayer TMDC moiré system provides an exciting new platform for excitonic physics and correlated physics. Considering the three high symmetric points (conventionally named as A, B C; arXiv:1907.12338) for each of the TMDC moiré bilayers that can potentially trap the electrons or holes, it will be interesting to explore how these high symmetry points from the top bilayer align with the bottom bilayer. With accurate angle control, it is possible to realize an array of these points with different combinations (such as AAA, ABA, or ABC, etc.), which ushers in unprecedented opportunities in electronic and excitonic band structure engineering and a new route to realizing new quantum states.

2. The interlayer and intralayer excitons in the bilayer WSe_2/WS_2 heterostructure are shown to be localized (Nature 603, 247–252 (2022), Nature 609, 52–57 (2022), Science 378, 1235-1239 (2022)). Are the excitons in the trilayer expected to be localized as well? What is the influence of the moiré potential and reconstruction on the quadrupolar exciton?

We thank the reviewer for the insightful question. In the extremely simplified scenario, where we envision the perfect alignment of the top and bottom layers (including the high symmetric points),

we envision that the holes, albeit shared between the top and bottom WSe₂ layer, are still localized laterally, thus the quadrupolar excitons should still be localized. However, one might envision that the high symmetry points are not perfectly aligned, either due to a finite twist angle between the top and bottom moiré bilayers or a lateral movement, depending on the lateral misalignment, the confinement of the quadrupolar excitons can be reduced, and eventually, in the extreme case, the quadrupolar excitons cannot be confined anymore. It will be intriguing to find out whether the quadrupolar excitons can still be formed with a finite twist angle. We do not have capability to address all these intriguing questions in the current work, but we hope that our work will inspire future investigations in this direction.

In terms of atomic reconstruction, we believe that from the moiré intralayer exciton resonances, both top and bottom bilayer moiré bilayers will have atomic reconstruction as recently reported (Nature Materials volume 20, 945–950 (2021)). In the trilayer region, in the scenario of perfect alignment, there will not be further reconstruction needed, as the top and bottom bilayers will have the same lattice constant. In the scenario of a small twist angle, there might be further reconstruction involved but we suspect that it will be a small effect as such reconstruction should occur over a much longer length scale (the new moiré lattice formed by the two slightly misaligned top and bottom moiré superlattices) moiré.

3. Why do Fig. 1 e and f, which are both for dipolar excitons, show quite different doping dependences?

The PL of the interlayer exciton involves both absorption and quantum yield. As a result, the PL is very sensitive to the local environment, different from the absorption/reflectance spectra. Since Region I and region II are tens of micrometers apart, we expect the interlayer exciton PL will not look exactly the same.

But we think the PL from Region I and Region II share mostly similar features in terms of their gate dependence: they both have strong intensity near charge-neutral region; the peak shift and integrated PL intensity all show the correlated states at $n=-1$ and 1.

Now we have replaced Fig.1 of the original main text with the data from device D5, shown also in Fig. R2. We believe the PL from Region I and II are more similar now, likely due to improved inhomogeneity.

4. The author should comment on how the purple spheres are extracted in Fig. 2(d).

The purple spheres (PL peak positions) in the original Fig. 2d were extracted by searching for local maxima of the PL spectra. In the revised manuscript, we show in the revised Fig.2d the fitted peak positions by fitting the PL spectra with a single Lorentzian peak.

Reviewers' Comments:

Reviewer #1:

Remarks to the Author:

After careful reading of the response letter and revised manuscript I conclude that the authors have adequately and clearly addressed all of my previous comments. I have no further questions, therefore I highly recommend this work for publication in Nature Communications. One minor, optional suggestion is to substitute "Electrical Field" with the widely used term "Electric Field" consistently throughout the manuscript.

Reviewer #2:

Remarks to the Author:

I'm appreciated of the authors' effort. The revised manuscript has addressed all my concerns, especially the explanation of situation related to the relevant works. The new Figure 1 is better and more convincing than the previous one. I would thus recommend the publication of the manuscript in Nature Communications.

Reviewer #3:

Remarks to the Author:

The authors have satisfactorily responded to all my comments. I recommend publication.

We thank all reviewers for their efforts again. We have addressed the remaining question in the reply below.

In addition, to comply with the format requirements of Nature Communications, We have moved the extended Figs.1, 2, 3 and 4 in the original manuscript into the revised Supplementary Information as Fig.S5, Fig.S6, Fig.S7 and Fig.S15.

With these revisions, we believe that we have addressed all reviewers' questions, and our manuscript is ready for the publication in Nature Commications. We thank all reviewers for helping us improve our manuscript.

REVIEWERS' COMMENTS

Reviewer #1 (Remarks to the Author):

After careful reading of the response letter and revised manuscript I conclude that the authors have adequately and clearly addressed all of my previous comments. I have no further questions, therefore I highly recommend this work for publication in Nature Communications. One minor, optional suggestion is to substitute "Electrical Field" with the widely used term "Electric Field" consistently throughout the manuscript.

We thank the reviewer for the recognition of our work and the recommendation for publication. We have revised the manuscript according to the reviewer's suggestion.

Reviewer #2 (Remarks to the Author):

I'm appreciated of the authors' effort. The revised manuscript has addressed all my concerns, especially the explanation of situation related to the relevant works. The new Figure 1 is better and more convincing than the previous one. I would thus recommend the publication of the manuscript in Nature Communications.

We thank the reviewer for the recognition of our work and the recommendation for publication.

Reviewer #3 (Remarks to the Author):

The authors have satisfactorily responded to all my comments. I recommend publication.

We thank the reviewer for the recognition of our work and the recommendation for publication.